# Oncolytic BHV-1 Is Sufficient to Induce Immunogenic Cell Death and Synergizes with Low-Dose Chemotherapy to Dampen Immunosuppressive T Regulatory Cells

**DOI:** 10.3390/cancers15041295

**Published:** 2023-02-17

**Authors:** Maria Eugenia Davola, Olga Cormier, Alyssa Vito, Nader El-Sayes, Susan Collins, Omar Salem, Spencer Revill, Kjetil Ask, Yonghong Wan, Karen Mossman

**Affiliations:** 1Department of Medicine, Centre for Discovery in Cancer Research, McMaster University, Hamilton, ON L8S 4K1, Canada; 2Firestone Institute for Respiratory Health, St. Joseph’s Healthcare Hamilton, Hamilton, ON L8N 4A6, Canada

**Keywords:** immunogenic cell death, oncolytic virus, bovine herpesvirus type 1, immune checkpoint inhibitors, mitomycin c, T regulatory cells

## Abstract

**Simple Summary:**

Immunotherapy is designed to stimulate the patient’s own immune system to fight their specific cancer. While immune checkpoint therapies work well for some tumors, they fail to work in tumors that are “immune cold”. Oncolytic viruses are viruses that preferentially target tumor cells while sparing healthy cells and can help stimulate immune responses. An oncolytic virus based on a common human herpesvirus has been granted FDA approval, and is currently being used as an intralesional cancer immunotherapy. We have previously shown that a related bovine herpesvirus has many unique properties that suggest widespread use against many types of primary and metastatic cancers. Here, we show for the first time in an immune competent mouse model that oncolytic BHV-1 can activate a type of immune response that can turn “cold” tumors “hot”. Addition of low-dose chemotherapy to oncolytic BHV-1 increases good immune responses while decreasing harmful immune responses, allowing immune checkpoint therapy to clear tumors.

**Abstract:**

Immunogenic cell death (ICD) can switch immunologically “cold” tumors “hot”, making them sensitive to immune checkpoint inhibitor (ICI) therapy. Many therapeutic platforms combine multiple modalities such as oncolytic viruses (OVs) and low-dose chemotherapy to induce ICD and improve prognostic outcomes. We previously detailed many unique properties of oncolytic bovine herpesvirus type 1 (oBHV) that suggest widespread clinical utility. Here, we show for the first time, the ability of oBHV monotherapy to induce bona fide ICD and tumor-specific activation of circulating CD8^+^ T cells in a syngeneic murine model of melanoma. The addition of low-dose mitomycin C (MMC) was necessary to fully synergize with ICI through early recruitment of CD8^+^ T cells and reduced infiltration of highly suppressive PD-1^+^ Tregs. Cytokine and gene expression analyses within treated tumors suggest that the addition of MMC to oBHV therapy shifts the immune response from predominantly anti-viral, as evidenced by a high level of interferon-stimulated genes, to one that stimulates myeloid cells, antigen presentation and adaptive processes. Collectively, these data provide mechanistic insights into how oBHV-mediated therapy modalities overcome immune suppressive tumor microenvironments to enable the efficacy of ICI therapy.

## 1. Introduction

Oncolytic viruses (OVs) are being actively studied as novel cancer therapeutics since they preferentially target and kill tumor cells and can stimulate an anti-tumor, patient-specific immune response [1]. Moreover, OVs can kill cancer stem cells and replicate in hypoxic environments and in drug-resistant cells [2,3]. While direct oncolysis was initially considered the primary mechanism of action of OVs, initiation of an anti-tumor immune response is now considered an essential aspect of OV therapy [4,5]. Indeed, OVs are potent inducers of in situ vaccination, using the tumor as the antigen source for immune stimulation [6,7,8]. OVs turn “immune cold” tumors “immune hot”, allowing tumors to effectively respond to immune checkpoint inhibitor (ICI) therapy [9]. It is believed that OV-mediated initiation of host anti-tumor immunity is a result of the induction of immunogenic cell death (ICD) [10,11]. ICD then triggers innate and adaptative anti-tumor immune activation [12].

Combination platforms using OVs with low-dose chemotherapy have shown increased ICD induction over monotherapies alone with increased infiltration of CD8^+^ T cells into tumors [13], therefore yielding greater potential for synergy with ICI [14,15,16]. Pre-clinical and clinical trials have validated OVs [17] with a herpes simplex virus type 1 (HSV-1)-based OV approved by the USA Food and Drug Administration (FDA) for the treatment of melanoma in 2015 [18]. However, clinical trials have also revealed the limitations of OVs [19]. The pre-existence of antibodies against human OVs or rapid seroconversion after OV administration accelerates virus neutralization, limiting viral delivery to the tumor [20]. Additionally, many OVs are restricted by mutations and/or expressed antigens in the tumor, limiting tumor targeting [19]. Thus, new strategies are required to overcome the current clinical challenges of OV-based therapies and improve their translative potential.

In this sense, bovine herpesvirus type 1 (BHV-1), a close relative of HSV-1, has exciting properties for clinical development. Unlike HSV-1, BHV-1 does not cause disease in humans or infect healthy human cells and is not neutralized by human serum [21], allowing for either intratumoral or intravenous delivery for the treatment of primary and metastatic cancers. In addition to targeting multiple cancer subtypes, similar to HSV-1, BHV-1 also targets immortalized cells, suggesting pre-neoplastic to overt cancer cell activity [21]. It further targets bulk and cancer-initiating tumor cells, regardless of tumor status or subtype [2]. Oncolytic BHV-1 expressing green fluorescent protein (oBHV) killed more human cancer cell lines from the NCI60 panel with greater killing capacity than an ICP0-null oncolytic HSV-1 [22]. Of clinical relevance, oBHV has a particular affinity for cancers with a high incidence of mutant KRAS [22], a negative therapeutic outcome predictor of hard-to-treat cancers such as lung, colorectal and pancreatic cancers. In the NCI60 panel, the multiplicity of infection (MOI) of oBHV used, but not replication level, correlated with the cell-killing capacity [22]. As with other non-human viruses being developed as OVs, a fulsome understanding of how BHV-1 behaves in vivo is critical to ensure patient safety.

In vivo studies of BHV-1 are limited due to the lack of syngeneic animal models [23,24,25]. While cotton rats are susceptible to BHV-1, they are a challenging model to work with and limited reagents are available [26]. Nevertheless, we have previously shown that oBHV, in combination with the methyltransferase inhibitor 5-Azacytidine, significantly decreased the incidence of secondary lesions with enhanced tumor cell clearance and evidence of immune cell infiltration in a tolerized cotton rat model of breast adenocarcinoma [27]. Standard murine models are unsuitable as murine cells lack BHV-1 entry receptors [24,25]. In this study, we used a modified B16 melanoma mouse model (B16-C10) [28] expressing nectin-1, a human receptor for BHV-1 entry [29,30], to further investigate oBHV anti-tumor immunostimulatory properties. We found that while oBHV alone is sufficient to induce bona fide ICD and induce the activation of circulating CD8^+^ T cells and tumor-infiltrating lymphocytes (TILs), low-dose mitomycin (MMC) chemotherapy is required to enable ICI to significantly extend survival.

## 2. Materials and Methods

### 2.1. Cell Lines

Cell lines were maintained at 37 °C with 5% CO_2_ in a medium supplemented with 1 mM L-glutamine. MDBK-derived CRIB cells [31] were obtained from Prof. Clinton Jones (Oklahoma State University, US) and cultured in Dulbecco’s modified Eagle’s medium (DMEM) supplemented with 5% fetal bovine serum (FBS). B16 control and B16-C10 cells [28], which are derived from B78H1 cells, were obtained from Prof. Gary Cohen (University of Pennsylvania, US) and maintained in DMEM supplemented with 5% FBS, 100 U/mL penicillin–streptomycin and 250 μg/mL Geneticin (Gibco Cat# 10131035).

### 2.2. Oncolytic Virus

oBHV, a BHV-1 recombinant that expresses green fluorescent protein (GFP), was a kind gift from Dr. Günther Keil (Friedrich-Loeffler-Institut, Greifswald, Mecklenburg-Vorpommern, Germany) [32]. oBHV contains murine cytomegalovirus (CMV) promoter and enhanced GFP sequence in the glycoprotein *gI* locus, resulting in *gI* disruption. oBHV was propagated and titrated on CRIB cells. The virus preparations included sucrose cushion purified, and the purified virus was resuspended in PBS and stored at −80 °C [2].

### 2.3. Drug Preparation

Mitomycin c (MMC) stock powder (Sigma-Aldrich, St. Louis, MO, USA Cat# M4287) was stored at 4 °C and dissolved in sterile water to a concentration of 2 mg/mL. MMC was freshly prepared for each experiment. α-CTLA-4 and α-PD-L1 antibodies (BioXCell Cat# BE0131 and BE0101, respectively) were diluted to 1 mg/mL with sterile PBS.

### 2.4. Tumor Regression Studies in Mice Bearing B16-C10 Tumors

Female C57Bl/6 mice were maintained at the McMaster University Central Animal Facility; procedures were performed in full compliance with the Canadian Council on Animal Care and approved by the Animal Research Ethics Board of McMaster University. Then, 6- to 8-week-old C57Bl/6 mice were subcutaneously implanted with 5 × 10^6^ B16-C10 cells resuspended in 200 µL PBS into the left flank [28]. Tumors reached palpable size (50–100 mm^3^) within 2 weeks of the injection (day 1). For MMC monotherapy, tumors were treated intratumorally (i.t.) with 100 µg MMC in 50 µL sterile water only at day 1. For oBHV monotherapy, tumors were treated i.t. by injecting 2 × 10^7^ pfu oBHV in 50 μL PBS once daily beginning at day 2 for three consecutive days (days 2, 3 and 4) [13,27]. For ICI monotherapy, tumors were treated with mouse α-CTLA-4 and α-PD-L1 antibodies intraperitoneally (i.p.) (200 µg each) from day 2 every 3 days until 10 doses total. For combinatorial therapies, tumors were treated with MMC or 50 µL sterile water i.t. at day 1, oBHV or 50 µL PBS i.t. at days 2, 3 and 4, and α-CTLA-4 and α-PD-L1 antibodies or 200 µL PBS from day 2 every 3 days until 10 doses total. Tumors were measured every 3–4 days using a digital caliper from day 1 until endpoint, when tumors reached a volume of 525 mm^3^.

### 2.5. Histology and Image Analysis

Treated and control tumors were resected and fixed in 10% formalin for 48 h and then transferred to 70% ethanol until histological processing. Tumor tissue was embedded in paraffin and 4-μm sections were prepared. Tissue sections were processed for immunohistochemistry (IHC) using Automated Leica Bond Rx stainer with Epitope Retrieval Buffer 2 (Leica, Wetzlar, Germany, AR9640). All antibodies were diluted in IHC/ISH Super Blocker (Leica, PV6199). Primary antibodies and working dilutions were as follows: α-CD4 (1:800; Ebio) and α-CD8a (1:1000; Ebio). Secondary antibody and working dilution: rabbit α-rat antibody (1:100; Vector Labs). The Bond Refine Polymer detection kit (Leica, Concord, ON, Canada) was used.

Staining was digitalized using the Olympus VS120-L100-W automated slide scanner. Slides were batch-scanned on the brightfield setting at 20× magnification. The color camera used was the Pike 505C VC50. HALO Image Analysis Software (Indica Labs, HALO v2.2) was used to analyze digital histology images. Cytonuclear cell count algorithms were developed to determine the amount of CD4 and CD8a positive cells and total cell numbers in a given tissue sample. The percentage of positive cells was calculated relative to total cell number.

### 2.6. Gold Standard ICD Immunization Assay

Female C57Bl/6 mice were vaccinated with dying B16-C10 cells. B16-C10 cells were mock- or infected with oBHV at MOI of 20 for 1 h at 37 °C; the virus inoculum was removed, cells were washed with PBS and media were added back. After 2 h at 37 °C, the media were replaced with fresh media with or without 20 µg/mL MMC and cells were incubated for 14 h at 37 °C. The optimal timepoint for harvesting cells that were in the process of dying was chosen as the timepoint where cellular viability started to decrease compared to untreated cells (see Appendix A, 14 h post-infection). Treated cells and supernatant were harvested and centrifugated at 1000 rpm for 10 min at 4 °C. Pellets were resuspended in PBS and used for vaccination. Each mouse was vaccinated subcutaneously with only PBS (N = 8) (as carrier control) or 3 × 10^6^ cells (200 µL) (N = 10) on the left flank. Three days post-vaccination, 5 × 10^6^ B16-C10 cells were implanted on the right flank (challenge). Once palpable, tumors were measured every 3–4 days until endpoint.

### 2.7. Tumor-Specific Activation of Circulating CD8^+^ T Cells

On day 7 post-treatment, blood was collected from the periorbital sinus of mice and red blood cells were lysed using ACK buffer. Peripheral blood mononuclear cells (PBMCs) were stimulated with a collection of tumor B16-associated peptides: gp70, gp100 and P15e (1 μg/mL) in the presence of brefeldin A (GolgiPlug; BD Pharmingen Cat# 555029, 1μg/mL added after 1 h of incubation). After 5 h of incubation, cells were treated with α-CD16/CD32 (Fc block; BD Pharmingen Cat# 553142) and the surface stained with α-CD8a antibody (BD Pharmingen Cat# 552877). Cells were then permeabilized and fixed with cytofix/cytoperm (BD Pharmingen Cat# 554714) and stained for intracellular interferon γ (IFNγ) (BD Biosciences, Franklin Lakes, NJ, USA, Cat# 554410). Data were acquired using LSRFortessa flow cytometer with FACSDiva software (BD Pharmingen) and analyzed with FlowJo Mac, version 10.6 software (BD, Ashland, OR, USA).

### 2.8. Re-Challenge Experiment

Mice that completely responded to triple combination therapy (tumor-free mice) (N = 4) and naïve 6- to 8-week-old control mice (N = 10) were subcutaneously implanted with 5 × 10^6^ B16-C10 cells resuspended in 200 µL PBS into the right flank (opposite of initial tumor cell implantation). Once palpable, tumors were measured every 3–4 days for 4 weeks.

### 2.9. Flow Cytometry Analysis

Tumors were minced with a razor blade in RPMI media. Then, 50 µg/mL liberase (Sigma Aldrich Cat# 5401054001) was added for digestion and samples incubated for 20 min at 37 °C. The cell suspension was passed over a 100-micron filter and rinsed with 5 mL of RPMI. Samples were spun at 1500 rpm for 5 min. Red blood cells were lysed using ACK buffer. The PBMCs were treated with anti-CD16/CD32 (Fc block; BD Biosciences Cat# 553141) and the surface stained with fluorescently conjugated antibodies for FVS (BD Biosciences, #564406), CD3 (BD Biosciences Cat# 564010), CD4 (BD Biosciences Cat# 561830), CD8 (BD Biosciences Cat# 563046), PD-1 (BD Biosciences Cat# 748266), Tim-3 (BD Biosciences Cat# 747626), CD25 (BD Biosciences Cat# 563061), and FoxP3 (BD Biosciences Cat# 560403). LSRFortessa flow cytometer with FACSDiva software (BD Biosciences) was used for data acquisition and FlowJo Mac, version 10.8.1 software, was used for data analysis.

### 2.10. Cytokine Analysis

B16-C10 tumors were treated with PBS, oBHV or MMC + oBHV as described previously. On days 3 and 5, tumors were resected and cut into small pieces and homogenized in the presence of a tissue extraction solution (50 mM Tris, pH 7.4, 250 mM NaCl, 5 mM EDTA, 2 mM Na_3_VO_4_, 1 mM NaF, 20 mM Na_4_P_2_O_7_, 1 mM beta-glycerophosphate, 1% NP-40) [14]. Homogenized tumors were incubated on ice for 30 min. Whole-tumor lysates were clarified using three sequential centrifugations at 14,000 rpm for 10 min at 4 °C. Tumor homogenates were diluted to achieve equal amounts of protein concentration. The 44-Plex murine cytokine/chemokine analysis was conducted by Eve Technologies (Calgary, AB, Canada).

### 2.11. Clariom S Assay Transcriptional Profiling

B16-C10 tumors were treated with PBS, oBHV or MMC + oBHV as described previously. Tumors were collected from mice on day 5 and homogenized in Trizol (Invitrogen, Life Technologies Corporation, Carlsbad, CA, USA). Following Trizol extraction, the RNA was further purified using the Qiagen RNA extraction kit (Cat #74004). Extracted RNA was diluted to a concentration of 100 ng/µL and underwent reverse transcription. sscDNA was purified using magnetic beads and fragmented using UDG. The fragmented sample was hybridized to the Affymetrix Clariom S mouse arrays, and the stained arrays were scanned to generate intensity data. All of the reagents for this assay were developed by and purchased from Thermo Fisher Scientific. Raw data were analyzed using the Thermo Fisher Transcriptome Analysis Console software, version 4.0.2.1.5. The complete dataset can be found in the GEO database, TBD.

### 2.12. Statistical Analysis

Student’s *t*-test was used to compare two groups of data, while one-way ANOVA was performed to determine the difference among three or more groups. Kaplan–Meier curves were used to estimate survival, and the log-rank Mantel–Cox test and the Gehan–Breslow–Wilcoxon test were used to determine the difference in survival. The null hypothesis was rejected for *p*-values less than 0.05. All data analyses were carried out using GraphPad Prism.

## 3. Results

### 3.1. Use of B16-C10 Syngeneic Melanoma Model for Pre-Clinical Analysis of oBHV

While OVs display limited therapeutic efficacy as monotherapies, several studies have shown that OVs synergize with low-dose chemotherapies [13,15] at doses that exhibit immunomodulatory but not cytotoxic properties [33,34,35]. MMC has been studied extensively in the context of ICD [36,37,38] and with OVs, including oncolytic herpesviruses [39,40,41,42]. Here, we evaluated the therapeutic efficacy of oBHV alone or in combination with low-dose MMC. To overcome the lack of syngeneic mouse models susceptible to oBHV, we used B16-C10 cells [28], genetically engineered B16 mouse melanoma cells expressing nectin-1, a human BHV-1 entry receptor [29]. B16-C10 cells form tumors reproducibly in C57Bl/6 mice, and nectin-1 expression does not induce detectable in vivo immunogenicity against tumors [28]. In vitro results confirm that nectin-1 expression improves oBHV entry into B16 cells (Appendix A). However, oBHV replication in B16-C10 cells was very low as shown in Appendix A, where it was compared to a highly productive OV, such as oncolytic HSV-1 (oHSV) [43].

C57Bl/6 mice bearing subcutaneous B16-C10 tumors were treated i.t. with PBS, three doses of oBHV (2 × 10^7^ pfu each), one dose of MMC (100 μg) or the combination of MMC and oBHV, as shown in Figure 1A. Tumor volumes were monitored every 3–4 days until the mice reached endpoint. Although neither oBHV nor MMC showed efficacy as monotherapies, their combination slightly slowed tumor growth (Figure 1B), though without a significant increase in survival (Figure 1C). The lack of efficacy of oBHV monotherapy is not related with its low-productive replication in vitro in B16-C10, as a highly productive oHSV also failed as monotherapy (Appendix A). Of interest, MMC failed to enhance oBHV low-productive replication in vitro. In contrast, MMC significantly dampened the production of de novo virus (Appendix A) but kept B16-C10 viability at levels comparable to those of MMC alone (Appendix A). To investigate the immune landscape of untreated and treated B16-C10 tumors, a histologic assessment was performed. Tumors were harvested on days 7 and 10 from mice treated with PBS, oBHV, MMC or MMC + oBHV, and stained with antibodies for CD4 and CD8a to assess the level of T cell infiltration. Only tumors treated with MMC + oBHV have statistically significant infiltration of CD4^+^ T cells into the tumor, as compared to PBS-treated tumors, at day 10 (Figure 1D). The CD8a^+^ infiltration showed a different profile (Figure 1E). While, at day 7, significantly higher CD8a^+^ TILs were seen in all treated tumors compared to PBS control, only tumors treated with MMC or MMC + oBHV maintained higher CD8a^+^ TIL levels at day 10 (Figure 1E).

### 3.2. oBHV Alone Is Sufficient to Induce ICD and Tumor-Specific CD8^+^ T Cell Activation

The mounting of a systemic anti-tumor immune response can be partially explained by the ability of therapies such as OVs to induce ICD of cancer cells [10,11]. The gold standard assay to assess bona fide ICD uses dying tumor cells as a vaccine to determine if the type of cell death is sufficient to induce an immune response capable of limiting or controlling subsequent tumor formation [44]. Our group has published that oncolytic HSV-1 requires low-dose MMC, a non-ICD-inducer per se [36], to induce bona fide ICD and recruit TILs [14,16]. To further understand if oBHV also needs MMC to induce ICD and trigger T cell infiltration into the tumor, dying B16-C10 cells infected with oBHV or infected and treated with MMC (MMC + oBHV) were used to vaccinate naïve mice (Figure 2A). Conditions were determined as being sufficient to initiate cell death (Appendix A). As a negative control of ICD, a group of mice received only PBS (carrier, no cells), and another group received cells that were treated only with MMC (non-ICD-inducer).

As expected, vaccination with dying cells treated with MMC alone fails to limit tumor progression compared to the PBS (no cell) control, confirming the inability of MMC to kill B16-C10 cells in an immunogenic manner (Figure 2B). In contrast, vaccination with dying cells infected with oBHV alone was as effective as MMC + oBHV in limiting tumor growth (Figure 2B). Additionally, both killing methods of B16-C10 cells showed comparable efficacy as infected-cell vaccines for the treatment of mice bearing B16-C10 tumors (Appendix A). These data suggest that oBHV alone is sufficient to induce bona fide ICD of B16-C10 cells. Of interest, while MMC + oBHV increases cell death in vitro compared to oBHV alone (2 days post-infection, Appendix A), it does not increase the immunogenicity of dying cells (Figure 2B).

**Figure 2 cancers-15-01295-f002:**
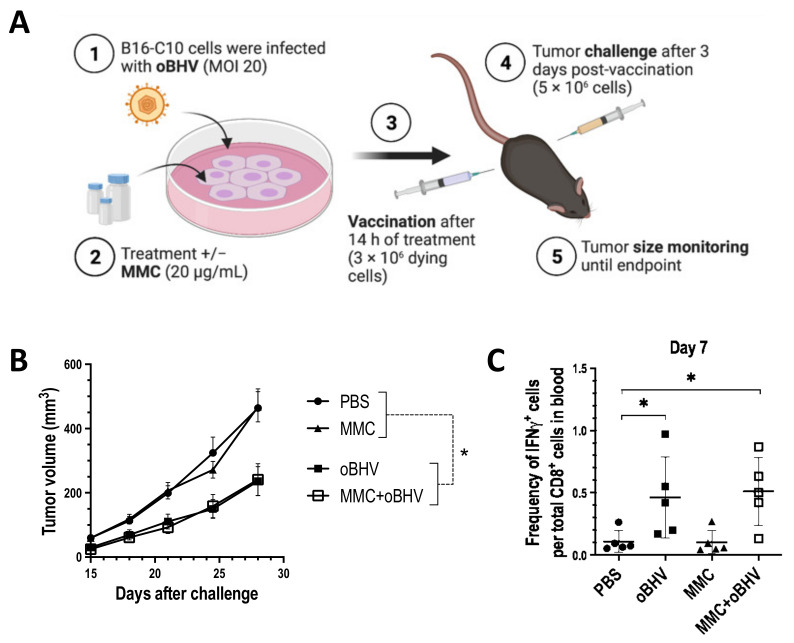
(**A**) B16-C10 cells were mock- or infected with oBHV (1) and, after 2 h, they were treated with or without MMC (2). After 14 h incubation, dying cells were used to vaccinate mice (3) and, 3 days later, untreated B16-C10 cells were implanted (challenge) (4). (5) Tumor volumes were measured from day 15 after challenge every 3–4 days until endpoint. (**B**) Tumors volume mean ± SEM. (**C**) On day 7, blood was collected for IFNγ intracellular staining of CD8+ T cells stimulated with tumor-specific peptides. * *p* < 0.05.

ICD releases immunogenic molecules and tumor-associated antigens (TAAs), activating antigen-presenting cells (APCs) leading to the activation and recruitment of TILs [45,46]. The activation of circulating CD8^+^ T cells in tumor-bearing mice treated with PBS, MMC, oBHV or MMC + oBHV was evaluated on day 7 via IFN-γ ICS of CD8^+^ T cells. Consistent with the bona fide ICD assay, only oBHV (with or without MMC), induced significant tumor-specific activation of circulating CD8^+^ T cells (Figure 2C).

### 3.3. Low-Dose MMC Synergizes with oBHV to Sensitize Tumors to ICI and Induce Long-Term Protective Immunity

We previously reported that the combination of low-dose MMC with oncolytic HSV-1 induced ICD and increased TILs, thus sensitizing tumors to ICI therapy [14,16]. Given that oBHV alone is sufficient to induce ICD and increase TILs, we hypothesized that oBHV monotherapy would sensitize tumors to ICI. Mice bearing B16-C10 tumors were treated with PBS, a mix of anti-PD-L1 and anti-CTLA-4 antibodies (ICI), oBHV + ICI, MMC + ICI, or the triple combination MMC + oBHV + ICI, as shown in Figure 3A. ICI alone failed to control tumor growth reflecting the immune “coldness” of the B16-C10 model (Figure 3B). While the combinations oBHV + ICI and MMC + ICI showed partial tumor control over PBS (Figure 3B), no significant increase in survival was observed (Figure 3C). The triple combination therapy, however, demonstrated enhanced tumor control (Figure 3B) and a significant increase in overall survival with 70% animal survival (Figure 3C) and 40% complete remission at day 49 after treatment (data not shown). Mice that fully responded to triple combination therapy (tumor-free mice) were re-challenged at day 56 via subcutaneous injection of B16-C10 cells into the opposite flank. All tumor-free mice rejected the second round of tumor engraftment (Figure 3D), suggesting a systemic long-term immune response against the tumor.

### 3.4. Therapeutic Efficacy of Triple Combination Correlates with Early Tumor Infiltration of CD8^+^ T Cells and Reduced Tumor Infiltration of Higly Suppresssive PD-1^+^ Treg Cells

To further investigate TIL distribution in treated mice, a histologic assessment was performed. Tumors treated with PBS, ICI, oBHV + ICI and the triple combination of MMC + oBHV + ICI were harvested at days 7 and 10 and stained with antibodies for CD8a and CD4. The localization of CD8a^+^ (Figure 4A) and CD4^+^ (Figure 5A) T cells was analysed, and their percentages were quantified (Figure 4B and Figure 5B, respectively). As shown in Figure 4A, the distribution of CD8a^+^ T cells in the tumor periphery on day 7 was clear after both ICI and oBHV + ICI treatments, but a visibly larger number of CD8a^+^ T cells that infiltrated deep into the tumor tissue was observed with the triple combination. Tumors analyzed from day 10 show that oBHV + ICI-treated mice had comparable levels of recruitment (periphery) and infiltration (core) of cytotoxic T cells compared to triple-combination-treated mice (Figure 4A). Quantification shows that ICI, oBHV + ICI and MMC + oBHV + ICI significantly increased the percentage of CD8a^+^ TILs compared to PBS at day 7, with the triple combination inducing the highest levels (Figure 4B). By day 10, tumors treated with oBHV + ICI had similar levels of CD8a^+^ TILs compared to those with triple combination treatment (Figure 4B).

In addition, blood was collected at day 7 for IFN-γ ICS of CD8^+^ T cells stimulated with tumor-specific peptides (Figure 4C), and directly compared with tumor volumes to correlate levels of CD8^+^ T cell activation with tumor growth (Figure 4D). Mice treated with oBHV + ICI showed a high level of activation of circulating CD8^+^ T cells, which was minimally impacted by the addition of MMC (Figure 4C). However, the addition of MMC was clearly necessary to control tumor growth (Figure 4D), with activated CD8^+^ T cell levels within individual mice (indicated as a through f) failing to correlate with tumor control (Figure 4C,D).

The lack of correlation between day 10 CD8a^+^ TILs and therapeutic efficacy could potentially be explained by differential cytotoxic T cell (Tc) exhaustion. Tumors treated with PBS, oBHV + ICI or triple combination were processed on day 10 and Tc were stained for two well-known exhaustion markers, PD-1 (Figure 4E) and TIM-3 (Figure 4F), for analysis via flow cytometry. However, no significant differences of Tc exhaustion were observed between treatments (Figure 4E,F). Of interest, both oBHV + ICI and triple combination were able to significantly reduce PD-1^+^ Tc levels (Figure 4E) compared to PBS control.

**Figure 4 cancers-15-01295-f004:**
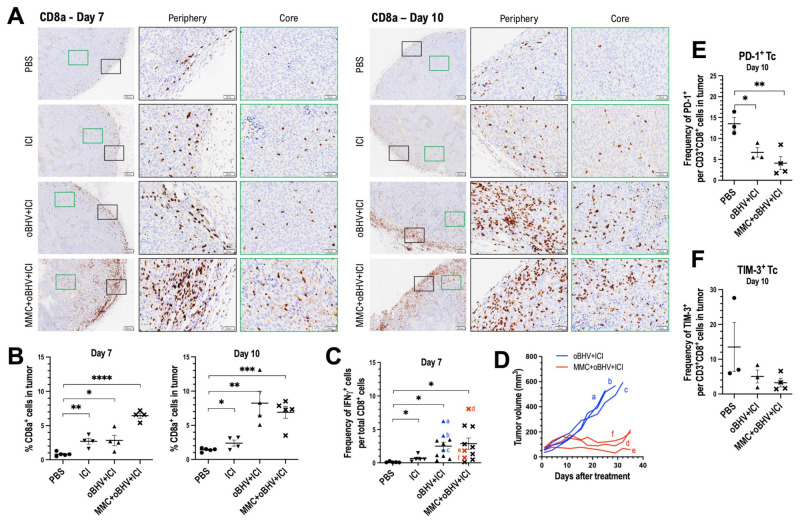
B16-C10 tumors were treated with PBS, ICI, oBHV + ICI or MMC + oBHV + ICI. CD8a IHC was performed on tumors on days 7 and 10 (**A**). Images with 2× magnification are shown in left panels, and 10× images of the tumor periphery (black) and core (green) are shown in the center and right panels. Scale bars: 200 μm (left panels); 50 μm (center and right panels). Quantification of CD8a^+^ cells was performed over the whole tumor area (**B**). On day 7, blood was collected for IFNγ ICS of peptide-stimulated CD8^+^ T cells (**C**). Mice shown as red and blue dots identified with letters a–f in C were monitored for tumor growth (**D**). To evaluate the level of exhaustion of CD8^+^ TILs (Tc) on day 10, tumors were processed and TILs were stained for analysis via flow cytometry (**E**,**F**). * *p* < 0.05, ** *p* < 0.01, *** *p* < 0.001, **** *p* < 0.0001.

Although CD8^+^ T cells are the most studied effector cells of cancer immunotherapy, CD4^+^ T cells are also required for mounting an efficient anti-tumor immune response. On day 7, while CD4^+^ T cells are observed in the periphery and core following all treatments (Figure 5A), the quantification indicated a significant increase only after combination treatments as compared to PBS-treated mice (Figure 5B). This significant increase in tumor-infiltrating CD4^+^ T cells was maintained at day 10 (Figure 5B). However, as shown in Figure 5A, a larger number of CD4^+^ T cells was found in the periphery and core of oBHV + ICI-treated tumors. Interestingly, the addition of MMC to oBHV + ICI treatment clearly decreased the number of CD4^+^ T cells both in the periphery and the core of tumors (Figure 5A).

To further investigate the difference in CD4^+^ TIL levels, treated tumors were harvested and processed on day 10, and CD4^+^ TILs were stained with CD25 and FoxP3 antibodies for flow cytometry analysis of Tregs. Triple combination therapy, but not oBHV + ICI, significantly reduced the proportion of CD25 ^+^ FoxP3^+^ Tregs among CD4^+^ T cells (Figure 5C) and increased the Tc:Treg ratio (Figure 5D). Only the triple combination treatment was able to reduce the proportion of highly suppressive PD-1^+^ Tregs (Figure 5E), while both combination treatments significantly reduced the TIM-3^+^ Treg population (Figure 5F).

**Figure 5 cancers-15-01295-f005:**
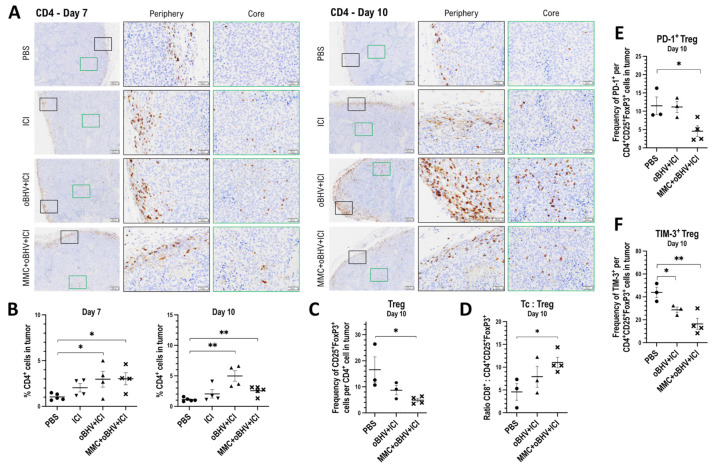
B16-C10 tumors were treated with PBS, ICI, oBHV + ICI or MMC + oBHV + ICI. CD4 IHC was performed on tumors on days 7 and 10 (**A**). Images with 2× magnification are shown in left panels, and 10× images of the tumor periphery (black) and core (green) are shown in the center and right panels. Scale bars: 200 μm (**left** panels); 50 μm (**center** and **right** panels). Quantification of CD4^+^ cells was performed over the whole tumor area (**B**). In addition, tumors were processed on day 10 and TILs were stained for analysis via flow cytometry to determine Treg TILs (**C**), ratio Tc: Treg (**D**) and level of activation of Treg TILs. * *p* < 0.05, ** *p* < 0.01.

### 3.5. Addition of MMC to oBHV Induces Early Release of Anti-Tumor Cytokines and Alters the Tumor Microenvironment

To better understand the role of MMC in combination therapy, tumors treated with PBS, oBHV or MMC + oBHV were harvested and processed for cytokine quantification at early timepoints (Figure 6A and Appendix A). One day after the administration of the first dose of oBHV (day 3), oBHV monotherapy induced the production of IL-6 and MCP-1, two pro-tumoral cytokines, LIF and VEGF, and the anti-tumoral cytokine IP-10. The addition of low-dose MMC increased the number of cytokines produced, including GM-CSF, TNFα, MIP-1a, MIP-1b and RANTES, five cytokines with reported anti-tumoral properties (Figure 6B). By day 5, monotherapy induced a wider array of cytokines with varied functions but mainly pro-tumoral, while combination therapy did not alter the number of cytokines produced, but their profile (Figure 6C).

To further explore the mechanism of action upon addition of MMC to oBHV therapy, an unbiased transcriptional profiling was conducted on B16-C10 tumors treated with PBS, oBHV or MMC + oBHV harvested at day 5 (Figure 6A and Figure 7). Unsurprisingly, oBHV strongly induces the expression of IFN-responsive genes (ISGs) as well as that of other immune stimulators, including known anti-viral responders. With addition of MMC, the immune response shifts relative to PBS-treated tumors. Notably, the induction of ISGs (Ifit3, Ifit3b, Ifit1, Ifi44, Oasl2, Usp18, Isg15 and Ddx58) is lower upon addition of MMC, while genes involved in other responses, such as Slfn4, Plac8, Ccl5 and Ly6c2, are upregulated to a greater extent (Figure 7A). Furthermore, the addition of MMC to oBHV induces a suite of gene-regulating processes, such as homeostasis, cell cycle progression, antigen presentation and the myeloid lineage, that are not induced in control or oBHV-treated tumors (Figure 7B). Of interest, the cytokine pattern at day 5 showed several cytokines upregulated by both oBHV and MMC + oBHV with reported ability to regulate myeloid cells (Figure 6C marked with *). These data suggest that MMC both modulates the inflammatory response induced by oBHV and synergizes with oBHV to induce new transcriptional responses which may contribute to the observed efficacy.

## 4. Discussion

While BHV-1 has previously been investigated as an OV [2,21,22,26,27,65,66], here, we validate the first immune competent murine cancer model for pre-clinical analysis of BHV-1. The use of immune competent animal models is imperative for studying OVs since the involvement of the host immune system is essential for OV therapy [4,5,45,46].

Despite immunotherapies showing promising results in clinical trials, one of the biggest limitations remains the highly immune suppressive nature of several cancer types. In this sense, induction of tumor ICD becomes incredibly important to reshape the tumor immune microenvironment by creating an inflammatory TME conducive for optimal antigen presentation and T cell activation [5,36,44,67,68,69]. In response to certain chemotherapies, UV radiation and OVs, tumors can undergo ICD [10,11]. We have previously reported that oncolytic HSV-1 (oHSV) requires combination with low-dose MMC, a non-ICD-inducer on its own, to induce bona fide ICD [14]. In addition, with oHSV, we consistently see a correlation between the induction of ICD, TILs and the therapeutic synergy of oHSV with ICI [13,15,16]. In contrast, we show here that oBHV alone is sufficient to induce ICD and activate tumor-specific CD8^+^ T cells in peripheral blood even when its in vitro replication is very low. The addition of MMC does not influence either event despite damping the de novo oBHV production in vitro and oBHV-mediated cell death being kinetically slower than MMC + oBHV treatment. While some tumors escape therapeutic pressures by downregulating the therapeutic target [70,71], given that oBHV replication is limited within treated B16-C10 cells, it is unlikely that oBHV-mediated downregulation of hNectin-1 accounts for the observed efficacy of combination therapy.

Regardless of oBHV inducing bona fide ICD, the addition of MMC is required for oBHV synergy with ICI. Histology of the immune TME revealed that both oBHV and MMC + oBHV combined with ICI induce similar recruitment of CD8^+^ TILs with similar activation and exhaustion levels by day 10, despite higher levels with triple combination at day 7. Histology of CD4^+^ TILs showed a different situation. Tumors treated with oBHV + ICI presented higher CD4^+^ TIL levels than triple combination; subsequent flow cytometry analysis revealed that only the triple combination significantly reduced the proportion of Tregs, increasing the Tc:Treg ratio. In particular, triple combination therapy decreases the proportion of highly suppressive PD-1^+^ Tregs. In accordance with ref. [72], these findings suggest a regulatory function of MMC in trafficking and activating Tregs in B16-C10 tumors. With that in mind, the insertion of different therapeutic entities into the virus backbone to provide functions that MMC provides, such as reprogramming Tregs into activated T helper cells [73,74,75], may be a strategic way to improve oBHV therapeutic efficacy and simplify the treatment modality.

To better understand what is happening in the TME after treatment with our combination strategies, we performed a cytokine analysis and transcriptional profiling. Neither the cytokine analysis nor the transcriptional profiling showed any pattern that can fully explain the correlation between the addition of MMC and Treg regulation. However, the cytokine pattern clearly showed a shift towards a more anti-tumoral TME upon addition of MMC, with upregulation of GM-CSF, TNF-α, MIP-1a, MIP-1b and RANTES at day 3, and IL-12p70, MIP-3b, Fractalkine and KC at day 5 [47,49,50,53,54]. Moreover, only tumors treated with oBHV monotherapy presented high levels of LIF. LIF can regulate CXCL9 in tumor-associated macrophages and prevent tumor infiltration of CD8^+^ T cells [48], explaining the slower CD8^+^ TILs kinetics seen with oBHV + ICI.

In addition, the cytokine pattern in the TME can potentially be used as biomarker of ICD [55]. OV-mediated ICD triggers the release of IL-6, IL-8, IFN-β, IP-10 (CXCL10), MCP-1 (CCL2) and KC (CXCL1) from cancer cells to recruit different immune cells and initiate an anti-tumor response [54,55,76,77]. As seen with the gold standard ICD assay, oBHV alone is sufficient to upregulate three ICD-related cytokines (IL-6, IP-10 and MCP-1). The addition of MMC only affects oBHV-induced IL-6 production at day 3 and upregulates KC at day 5. In contrast to our findings, IL-6 participates in the differentiation of naïve CD4^+^ T cells into Th17 while inhibiting their orientation toward the Tregs lineage [78], and increased levels of KC are associated with the recruitment of Treg to tumors [79]. However, both cytokines have pleiotropic properties: IL-6 is also related with pro-tumoral functions such as activation of carcinogenesis and tumor outgrowth, mediation of cytokine release syndrome and promotion of cachexia [47], and KC helps to recruit other immune cells, such as T cells and neutrophils, that can contribute to tumor control [80].

The transcriptional profiling showed that oBHV, as expected, induces a strong upregulation of several ISGs and other genes involved in the host anti-viral response. The addition of MMC causes a downregulation of some oBHV-upregulated ISGs (Ifit3, Ifit3b, Ifit1, Ifi44, Oasl2, Usp18, Isg15 and Ddx58) and an upregulation of genes involved in different pathways, including myeloid cell regulation (Slfn4, Ccl5, Ly6c2, Irgm1). Moreover, we found that only the combination MMC + oBHV upregulates another set of genes involved in myeloid pathways (Serpinb2, Ccl3, Ly6c1, Ms4a4a, Ccl9, Cd80, Sirb1a and Ms4a6c). Myeloid cells are among the most important defenders against infection, but they are also essential in tissue homeostasis and regulating T cell immunity [81]. In the context of cancer, myeloid cells have a controversial role in modulating tumor responses to various treatments [82]. In the case of oncolytic virotherapy, it has been reported that several OVs, including HSV-1, stimulate the release of a plethora of pro-inflammatory cytokines to recruit and activate myeloid cells and their differentiation into inflammatory (M1) macrophages or dendritic cells which engulf dying malignant cells and cross-present TAAs to naïve T cells [55]. Our cytokine analysis and transcriptional profiling also reflect the important role of myeloid cells in oBHV’s mechanism of action.

## 5. Conclusions

This study describes the first pre-clinical immune competent mouse model of cancer to evaluate the efficacy of oncolytic BHV-1. Despite being an alpha-herpesvirus akin to HSV-1, BHV-1 has many distinct properties that suggest its widespread and efficacious use as an oncolytic virus. Here, we show that oBHV is sufficient to induce bona fide ICD within immune cold melanoma tumors, resulting in the influx of CD4^+^ and CD8^+^ TILs. the addition of low-dose MMC chemotherapy does not enhance the immunogenicity of the treatment per se, distinct from its role with oHSV-1, but appears to shift the immune response from predominantly anti-viral, as evidenced by a high level of ISGs, to one that stimulates myeloid cells, antigen presentation and adaptive processes instead. This shift in the tumor microenvironment is critical to sensitize tumors to treatment with ICI, in part through the dampening of suppressive Tregs. These findings will enable the development of oBHV vectors expressing strategic transgenes to optimize clinical efficacy.

## Figures and Tables

**Figure 1 cancers-15-01295-f001:**
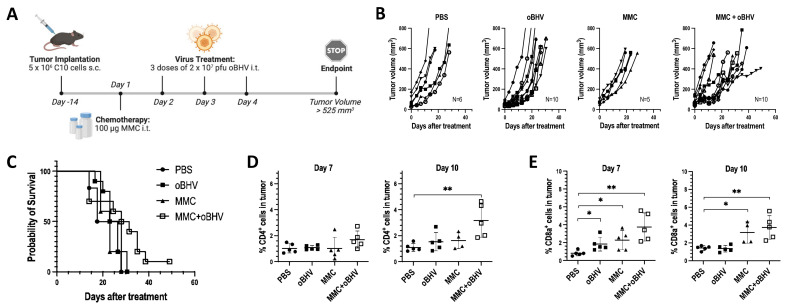
B16-C10 tumors were treated with PBS, oBHV, MMC or combination of MMC and oBHV (MMC + oBHV) (**A**). Tumors were measured every 3–4 days until animals reached endpoint. Tumor volume progression of each mice in each group (**B**) and Kaplan–Meier survival curve (**C**) are shown. Tumors were harvested on days 7 and 10 and histologic assessment with antibodies for CD4 and CD8a was performed to assess the level of T cell infiltration across the treatment groups. Quantification of CD4 (**D**) and CD8a (**E**) stains were performed over the whole tumor area. * *p* < 0.05 and ** *p* < 0.01.

**Figure 3 cancers-15-01295-f003:**
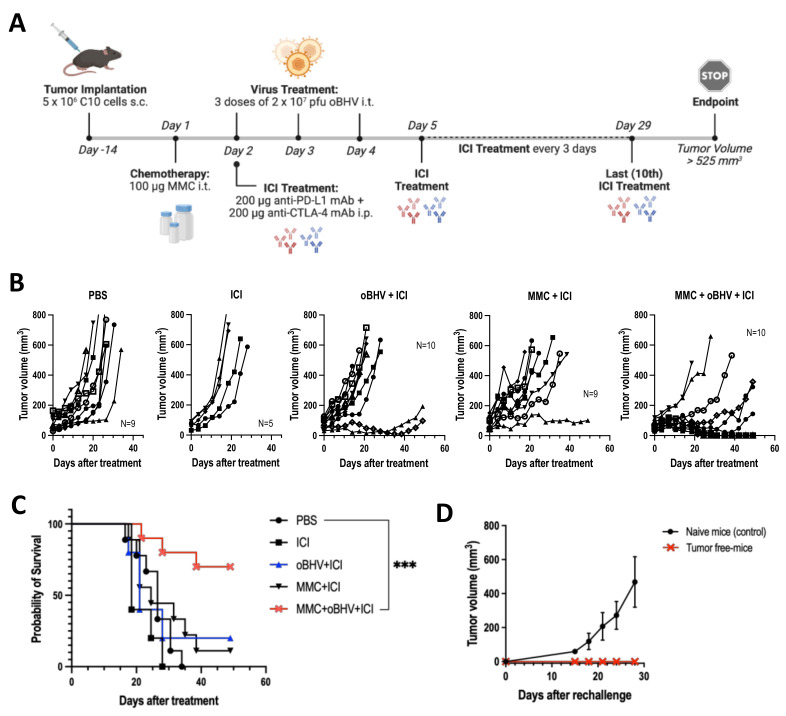
Tumors were treated with PBS, ICI (αCTLA-4 & αPD-L1) or different combinations of MMC, oBHV and ICI (**A**). Tumors were measured every 3–4 days until animals reached endpoint. Tumor volume progression of each mice in each group (**B**) and Kaplan-Meier survival curve (**C**) are shown. (**D**) Tumor-free mice that completely responded to MMC + oBHV + ICI treatment and naïve mice (control) were re-challenged by B16-C10 implantation. Tumors were measured every 3–4 days for 4 weeks. Tumor volume mean ± SEM is shown (**D**). *** *p* < 0.001.

**Figure 6 cancers-15-01295-f006:**
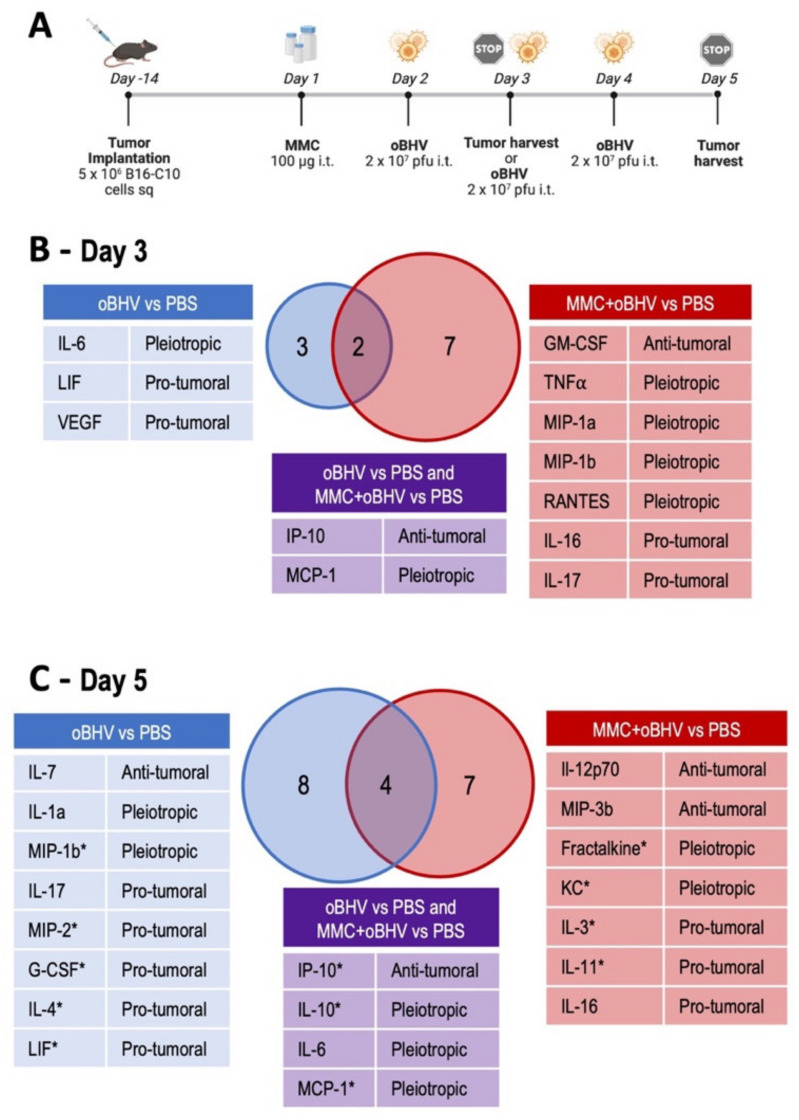
Cytokine analysis of tumors treated with PBS, oBHV or MMC + oBHV. On days 3 and 5, tumors were harvested and processing for cytokine quantification via Mouse Cytokine 44-Plex Discovery Assay (**A**). Venn diagrams represent the number of cytokines in treated tumors on day 3 (**B**) and day 5 (**C**) whose levels were significantly increased compared to PBS-treated tumors (*p* < 0.05). Each upregulated cytokine is shown with its reported function in cancer [47,48,49,50,51,52,53,54,55]. * indicates the cytokines that were reported to regulate production, activation or recruitment of myeloid cells [56,57,58,59,60,61,62,63,64]. MIP-1a = CCL3, MIP-1b = CCL4, RANTES = CCL5, IP-10 = CXCL10, MCP-1 = CCL2, Fractalkine = CX3CL1, MIP-2 = CXCL-2, MIP-3b = CCL19, KC = CXCL1.

**Figure 7 cancers-15-01295-f007:**
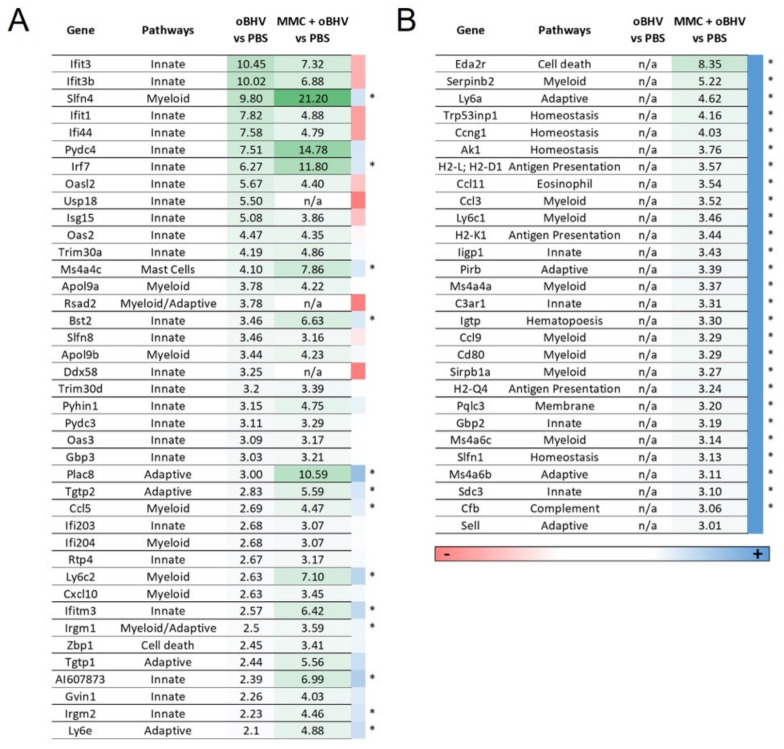
Transcriptional profiling of B16-C10 tumors treated with PBS, oBHV or MMC + oBHV showing trends in changes in expression with treatment. (**A**) Genes significantly upregulated by oBHV expressed as fold expression over PBS, compared to observed fold changes in MMC + oBHV treatment. (**B**) Genes significantly upregulated by MMC + oBHV expressed as fold expression over PBS that are not upregulated by oBHV alone. Genes suppressed or enhanced by addition of MMC indicated via color scale (red—suppressed, blue—enhanced). * indicates significant difference when signal data is compared directly (*p* < 0.05).

## Data Availability

Clariom S assay data (Figure 7) can be found in the GEO database (GSE223637).

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
