# Peer review of "Oncolytic BHV-1 Is Sufficient to Induce Immunogenic Cell Death and Synergizes with Low-Dose Chemotherapy to Dampen Immunosuppressive T Regulatory Cells"

_cancers, 2023, doi:10.3390/cancers15041295_

Round 1

Reviewer 1 Report

Davola et al. investigated the oncolytic activity of BHV-1 as monotherapy and in combination with mitomycin C chemotherapy and anti-PD-L1 and anti-CTLA-4 antibody immune checkpoint inhibitors (ICI) in a preclinical immunocompetent mouse model of melanoma cancer. In addition, the mechanisms of tumour microenvironment (TME) immunomodulation were investigated. The study shows that BHV monotherapy is not effective in targeting the cancer, while combination with mitomycin shows some efficacy and triple combination therapy has a significant therapeutic effect. Treatment with BHV-1 led to significant changes in the TME by inducing immunogenic cell death. Mitomycin chemotherapy did not enhance immunogenicity, but was shown to be important in sensitising the tumour to ICI.  

The study is well written and important mechanisms for modulation of TME by BHV-1 monotherapy and triple combination therapy were identified. 

Points to consider

1. unfortunately, BHV-1 monotherapy was not effective in vivo, but the reasons for this are not clear. One reason could be the low replication of the virus in the melanoma cell line B16-10. Supplementary Figure 2 showed that BHV-1 can infect B16-10 melanoma cells, but it is not clear whether the virus replicates in the cells and generates new virus particles that can infect non-infected cells. Detection of GFP, which is expressed by BHV-1 and has been shown to be an indicator of virus uptake and the start of replication, is not sufficient to answer this question. To confirm replication of the virus and to estimate the strength of virus replication in B16-10 cells, growth curves should be performed in B16-10 cells and (for comparison) in a highly susceptible cell line (time- and dose-dependent).

2. All experiments in which BHV-1 was combined with mitomycin C showed an improvement in therapeutic effect in vivo. Although several studies have been performed to clarify the reasons for this, I miss studies on the influence of mitomycin C on BHV-1 replication and BHV-1-induced cytotoxicity. Does mitomycin enhance BHV-1 replication and increase cell toxicity in the combination approach? Growth curves and cell viability tests in a combination approach in vitro could help to answer these questions.

Author Response

We thank this reviewer for his/her careful read of the manuscript and their thoughtful and constructive comments.  Our responses are indicated below: 

  1. unfortunately, BHV-1 monotherapy was not effective in vivo, but the reasons for this are not clear. One reason could be the low replication of the virus in the melanoma cell line B16-10. Supplementary Figure 2 showed that BHV-1 can infect B16-10 melanoma cells, but it is not clear whether the virus replicates in the cells and generates new virus particles that can infect non-infected cells. Detection of GFP, which is expressed by BHV-1 and has been shown to be an indicator of virus uptake and the start of replication, is not sufficient to answer this question. To confirm replication of the virus and to estimate the strength of virus replication in B16-10 cells, growth curves should be performed in B16-10 cells and (for comparison) in a highly susceptible cell line (time- and dose-dependent).

We use bovine CRIB cells for virus preparation and plaque assay as they are highly permissive, and we routinely obtain stock titres of 10e9 per mL.  As seen in new Figure S3A, oBHV replication in B16-C10 cells is low to negligible, particularly compared to oncolytic HSV-1 (described in lines 242-244 and 464-466). Regardless of enhanced replication of oHSV in B16-C10 cells, there is no therapeutic benefit as a monotherapy in vivo (new Figure S3B; described in lines 250-253). These data are consistent with other publications indicating herpesvirus replication in vitro does not routinely correlate with efficacy in vivo (summarized in Davola et al. 2019).

  1. All experiments in which BHV-1 was combined with mitomycin C showed an improvement in therapeutic effect in vivo. Although several studies have been performed to clarify the reasons for this, I miss studies on the influence of mitomycin C on BHV-1 replication and BHV-1-induced cytotoxicity. Does mitomycin enhance BHV-1 replication and increase cell toxicity in the combination approach? Growth curves and cell viability tests in a combination approach in vitro could help to answer these questions.

We have added Figure S4 (lines 253-256 and 466-467) to address this question. Of interest, mitomycin c fails to enhance oBHV low-productive replication. On the contrary, mitomycin c significantly dampens the production of new oBHV particles. The combination MMC+oBHV reduces B16-C10 viability at levels comparable to MMC alone. Along with our Clariom data, these data suggest that low level mitomycin C is influencing the tumor microenvironment in a mechanism distinct from enhancement of virus replication.

Reviewer 2 Report

Comments on manuscript cancers-2199386, entitled:

“Oncolytic BHV-1 is sufficient to induce immunogenic cell death and synergizes with low dose chemotherapy to dampen immunosuppressive T regulatory cells”

The authors show in this manuscript that oncolytic BHV-1 can induce immunogenic cell death in the tumor and can activate circulating  CD8+ T cells in a melanoma mouse model. The addition of a low dose of mitomycin C proved to be essential for the reduction of suppressive PD-1+ Tregs and sensitizes the tumors to immune checkpoint inhibitor therapy. The combination of oBHV, MMC and ICI, protected the treated animals against tumor growth after a re-challenge experiment, proofing a long-term immunity against the dying tumor cells. The manuscript is well-written and the conclusions are supported by the results.

 -          Paragraph 3.1. Use of B16-C10 syngeneic melanoma model for pre-clinical analysis of oBHV The authors tested the efficacy of oBHV-1 in a murine model with B16-C10 cells. These cells are modified to support BHV-1 infection and replication. It seems that treatment with the combination of MMC and oBHV does not significantly increase survival. Did the authors check if the tumor cells that did grow out in the oBHV-treated or oBHV-MMC-treated mice still contain the nectin-1 receptor? In a paper on reovirus in combination with bispecific antibodies in a mouse model with KPC3.TRP1 tumor cells, the tumor cells that eventually escaped the combination therapy,  no longer expressed TRP-1, making them resistant to the bispecific antibody treatment (Groeneveldt, C. J Immunother Cancer 2020, 8, (2), 10.1136/jitc-2020-001191).

 -          The authors show that the oBHV is very effective in a model with overexpression of nectin-1 in the tumor cells to achieve an effective infection and subsequent replication of the virus. I miss in the discussion the translation to human cancer cells, which maybe are not as sensitive for oBHV-1 or the pathways that are needed for the desired immune responses. What is the expectation from the authors on this?

-          BHV-1 is a pathogen of cattle and can (like some human herpes viruses) exists in a life-long latency state in cows/calves. Do the authors think that this can give problems for the use of the virus as an oncolytic virus? (Just out of curiosity)

 -          Paragraph 2, line 97. Cell lines. Could the authors provide a reference for the used CRIB cells? Similar question for line 105. Oncolytic Virus. Could the authors provide a reference for the oHBV virus containing the GFP sequence?

 -          GFP protein by itself is immunogenic and according to a recent paper in “Cancer cell”  by Grzelak et al. this protein may attribute to the immune responses measured in immune-competent mice (10.1016/j.ccell.2021.11.004). Could the addition of GFP in oBHV be partially responsible for the immune responses in the mice?

-          Paragraph 2, lines 116-129. Tumor regression studies in mice bearing B16-C10 tumors. There is no mention of the group sizes used for the experiments, could the authors provide this information in the paragraph? Also for paragraph 2, lines 171-175. Re-challenge experiment. How many mice were used in the groups for this experiment?

-          Figure 4B and C. The asterisks above the “significance” lines are lost.

-          Legend of Figure 7, line 416. “Enhaced” should be enhanced.

Author Response

We thank this reviewer for his/her thoughtful and constructive comments. Our responses are indicated below:

 -          Paragraph 3.1. Use of B16-C10 syngeneic melanoma model for pre-clinical analysis of oBHV The authors tested the efficacy of oBHV-1 in a murine model with B16-C10 cells. These cells are modified to support BHV-1 infection and replication. It seems that treatment with the combination of MMC and oBHV does not significantly increase survival. Did the authors check if the tumor cells that did grow out in the oBHV-treated or oBHV-MMC-treated mice still contain the nectin-1 receptor? In a paper on reovirus in combination with bispecific antibodies in a mouse model with KPC3.TRP1 tumor cells, the tumor cells that eventually escaped the combination therapy,  no longer expressed TRP-1, making them resistant to the bispecific antibody treatment (Groeneveldt, C. J Immunother Cancer 2020, 8, (2), 10.1136/jitc-2020-001191).

Thank you for pointing out this interesting finding. We have not checked the presence of nectin-1 in tumors after treatment. However, the selective loss of nectin-1 expression due to oBHV replication pressure is unlikely, given the low level of oBHV replication in B16-C10 cells (please see new Figures S3 and S4). oBHV contribution to the combinatorial treatment is more related with its ability to trigger an immune response and modify the TME than its onco “lytic” properties. Regardless, loss of nectin-1 remains a potential outcome of treatment, which we have added to the discussion (lines 468-473).

 -          The authors show that the oBHV is very effective in a model with overexpression of nectin-1 in the tumor cells to achieve an effective infection and subsequent replication of the virus. I miss in the discussion the translation to human cancer cells, which maybe are not as sensitive for oBHV-1 or the pathways that are needed for the desired immune responses. What is the expectation from the authors on this?

The capacity of oBHV to kill B16-C10 cells is low with low to negligible production of de novo virus (new Figures S3 and S4). Our studies using the NCI-60 panel of human cancer cell lines showed that oBHV targets 72% of the cell lines with varied killing capacity and replication, which is discussed in lines 74-80 (Cuddington etal. 2014).

-          BHV-1 is a pathogen of cattle and can (like some human herpes viruses) exists in a life-long latency state in cows/calves. Do the authors think that this can give problems for the use of the virus as an oncolytic virus? (Just out of curiosity)

To our knowledge, no reports of disease in humans have been reported and no studies of oBHV latency or shedding in humans have been published to date. Our earlier publications on oBHV also show that normal human cells are non-permissive to BHV-1. Moreover, similar to HSV-1, for which human cells are permissive, anti-herpetic drugs exist that could be used. We have added lines 80-81.

 -          Paragraph 2, line 97. Cell lines. Could the authors provide a reference for the used CRIB cells? Similar question for line 105. Oncolytic Virus. Could the authors provide a reference for the oHBV virus containing the GFP sequence?

References were added (lines 111 and 119).

 -          GFP protein by itself is immunogenic and according to a recent paper in “Cancer cell”  by Grzelak et al. this protein may attribute to the immune responses measured in immune-competent mice (10.1016/j.ccell.2021.11.004). Could the addition of GFP in oBHV be partially responsible for the immune responses in the mice?

Unfortunately, we do not have the oBHV homolog lacking GFP to perform the precise comparison with oBHV expressing GFP. However, we have performed in vivo experiments in the B16-C10 model using wild type BHV-1 and another BHV-1 recombinant expressing a different fluorescent protein and both OVs showed significant improvement in tumor control and animal survival when combined with MMC and ICI. Thus, while we cannot rule out the possibility that GFP contributes to the immunogenicity of oBHV, our data with wild type BHV-1 suggests any contribution is minimal.

-          Paragraph 2, lines 116-129. Tumor regression studies in mice bearing B16-C10 tumors. There is no mention of the group sizes used for the experiments, could the authors provide this information in the paragraph? Also for paragraph 2, lines 171-175. Re-challenge experiment. How many mice were used in the groups for this experiment?

For tumor regression studies, group sizes were added to the corresponding figures (Figures 1B and 3B). We use N between 5-10. In re-challenge experiment, 10 naïve mice and 4 tumor-free mice were used (added in M&M, see line 187). Group sizes for gold standard ICD assay was also added to the corresponding M&M section (line 170).

-          Figure 4B and C. The asterisks above the “significance” lines are lost.

It is possible that the images have lost resolution when they were pasted in the word document. Please see Figure 4.jpg image that was sent in a separate compressed folder named “Figures”.

-          Legend of Figure 7, line 416. “Enhaced” should be enhanced.

Legend of Fig 7 was updated (line 447).

Reviewer 3 Report

Davola and colleagues investigate an oncolytic bovine herpesvirus in an immune competent mouse model, with and without low dose MMC treatment and with and without ICI, using a mixture of anti-PD-L1 and anti-CTLA-4 antibodies. They study tumor growth inhibition, induction of immunity protecting against tumor rechallenge, T cell tumor infiltration, activation of circulating CTLs against TAAs and transcriptomic changes and cytokine release in the TME.

The reported observations are interesting. A couple of issues need to be addressed:

Please rephrase the statement on lines 16-17 in the abstract that the use of oncolytic HSV is restricted. While obviously the FDA-approved oncolytic HSV can only be used for treatment of the indication the virus was approved for, the statement suggests that HSV has a narrow field of potential application. In particular in view of the next sentence on lines 17-18 heralding the widespread use of bovine herpesvirus, the statement is misleading. Similarly, the sentence on lines 69-72, where advantages of BHV over HSV are listed, suggests that HSV cannot infect multiple cancer subtypes. Please rephrase to avoid misleading readers.

It is advised to present a more balanced review of the pros and cons of using BHV-derived oncolytic viruses in the introduction section. Currently, only advantages are listed. It would be fair to also mention potential disadvantages, such as e.g. concerns of introducing non-human viruses in humans.

The oncolytic virus oBHV that was used in all experiments was a gift from another group. No publication reference is given. If this is the first written disclosure of the virus, more detailed information about its construction should be given. The present description on lines 105-107 is too brief.

Tumor challenge and rechallenge were both given into the right flank (lines 157 and 174), while it is stated (lines 174-175) that rechallenge was done into the opposite flank.

Please explain or rephrase what is meant with “the combination partially slowed tumor growth” (line 232) in the Figure 1 experiment. Apparently, something else is meant here than the partial effect on tumor control described for the Figure 3 experiment (line 293). There, some of the tumors in the treatment group showed a clear effect compared to the control group whereas others did not. In Figure 1, no such effect is evident.

Please define “synergy” on line 438. Probably, this refers to synergy in tumor growth inhibition, not specific aspects of ICD, T cell recruitment etc.

The manuscript includes many self-citations; I count at least 16. This seems more than necessary, in particular because regularly multiple self-citations are used to support a single statement or are given where a reference does not seem needed at all.

In the Figure 1 legend, why is **** p<0.001 included? None of the comparisons shown has this significance.

In Figure 1A and 3A, subcutaneous is abbreviated as sq. (The abbreviation for square or sequence.) Please change to s.c.

In Figures 4B and 4E, the asterisks are missing.

Please check if the indicated significances in Figure 4C are correct.

Line 262: “As excepted” = “As expected”

Author Response

We thank this reviewer for his/her thoughtful and constructive comments.  Our responses are indicated, below:

Please rephrase the statement on lines 16-17 in the abstract that the use of oncolytic HSV is restricted. While obviously the FDA-approved oncolytic HSV can only be used for treatment of the indication the virus was approved for, the statement suggests that HSV has a narrow field of potential application. In particular in view of the next sentence on lines 17-18 heralding the widespread use of bovine herpesvirus, the statement is misleading. Similarly, the sentence on lines 69-72, where advantages of BHV over HSV are listed, suggests that HSV cannot infect multiple cancer subtypes. Please rephrase to avoid misleading readers.

It is advised to present a more balanced review of the pros and cons of using BHV-derived oncolytic viruses in the introduction section. Currently, only advantages are listed. It would be fair to also mention potential disadvantages, such as e.g. concerns of introducing non-human viruses in humans.

We thank this reviewer for these constructive comments.  We have modified the introduction to provide a more balanced view that represents the facts without misleading the readers (lines 15-17, 71-72 and 75-76).

The oncolytic virus oBHV that was used in all experiments was a gift from another group. No publication reference is given. If this is the first written disclosure of the virus, more detailed information about its construction should be given. The present description on lines 105-107 is too brief.

The appropriate reference was added (line 119).

Tumor challenge and rechallenge were both given into the right flank (lines 157 and 174), while it is stated (lines 174-175) that rechallenge was done into the opposite flank.

For tumor regression studies, tumors were implanted into the left flank (line 134). Mice with complete tumor regression (tumor-free mice) and naïve mice as control were re-challenged by implantation into the right (opposite) flank (lines 188). The challenge described on line 157, refers to the ICD assay. For ICD assay, mice were “vaccinated” into the left flank (line 170) and challenged into the right flank (lines 171).

Please explain or rephrase what is meant with “the combination partially slowed tumor growth” (line 232) in the Figure 1 experiment. Apparently, something else is meant here than the partial effect on tumor control described for the Figure 3 experiment (line 293). There, some of the tumors in the treatment group showed a clear effect compared to the control group whereas others did not. In Figure 1, no such effect is evident.

Thanks for pointing this out. We agree with your comment. We modified the sentence with “the combination slightly slowed tumor growth” (line 249).

Please define “synergy” on line 438. Probably, this refers to synergy in tumor growth inhibition, not specific aspects of ICD, T cell recruitment etc.

The “synergy” on lines 446 and 452 refers to synergy in tumor growth inhibition and animal survival improvement. We have clarified this within the text (lines 463 and 475).

The manuscript includes many self-citations; I count at least 16. This seems more than necessary, in particular because regularly multiple self-citations are used to support a single statement or are given where a reference does not seem needed at all.

We appreciate this point by the reviewer. However, we are one of the only groups that have developed BHV-1 as an oncolytic virus and refer to relevant studies that we have completed with oncolytic HSV and immunogenic cell death.  We endeavor in our manuscripts to provide the most relevant citations and to ensure all scientific points are cited.

In the Figure 1 legend, why is **** p<0.001 included? None of the comparisons shown has this significance.

Figure 1 legend was updated (line 271). Thank you for noting this error.

In Figure 1A and 3A, subcutaneous is abbreviated as sq. (The abbreviation for square or sequence.) Please change to s.c.

Figure 1A and 3A were edited. Thank you for noting this error.

In Figures 4B and 4E, the asterisks are missing.

It is possible that the images have lost resolution when they were pasted in the word document. Please see Figure 4.jpg image that was sent in a separate compressed folder named “Figures”.

Please check if the indicated significances in Figure 4C are correct.

Significances in Figure 4C were checked.

Line 262: “As excepted” = “As expected”

Text was edited (line 285). Thank you for noting this error.

Round 2

Reviewer 1 Report

My questions were answered sufficiently.